# Presumptive Non-Ischemic Priapism in a Cat

**DOI:** 10.3390/vetsci9010029

**Published:** 2022-01-14

**Authors:** Jeong-Min Lee, Ah-Won Sung, Han-Joon Lee, Joong-Hyun Song, Kun-Ho Song

**Affiliations:** College of Veterinary Medicine, Chungnam National University, Daejeon 34134, Korea; memo76777@gmail.com (J.-M.L.); aoooone@gmail.com (A.-W.S.); hanjoon1102@hanmail.net (H.-J.L.); jh.song@cnu.ac.kr (J.-H.S.)

**Keywords:** non-ischemic priapism, erection, penile trauma, cat

## Abstract

A 14-year-old neutered male British shorthair cat presented with a 21-day history of persistent erection and dysuria, along with overgrooming of the perineal region. Mild palpation induced pain and rigid corpora cavernosa with flaccid glans were observed during physical examination. Ultrasonography of the penis did not detect significant blood flow in the penile cavernosal artery. The drawing of aspirate blood from cavernosal bodies for gas analysis was impossible because of the anatomically small penis size of cats. Conservative management, including topical steroid ointment, lidocaine gel, gabapentin, and diazepam, was prescribed for supportive management. The clinical signs resolved, and ultrasonographic examination of the penis revealed no abnormalities. The cat remains clinically well without recurrence during the 6 months after treatment. To our knowledge, this is the first report of non-ischemic priapism in a cat.

## 1. Introduction

Priapism is a penile erection lasting more than 4 h in the absence of sexual stimulation and can be classified into non-ischemic (high-flow, arterial), ischemic (low-flow, veno-occlusive), and stuttering (recurrent, intermittent) subtypes [1,2,3,4,5]. Ischemic priapism is a medical emergency associated with acidosis, hypoxemia, and glucopenia within the corpus cavernosum, while the non-ischemic kind is a persistent erection resulting from unregulated arterial inflow, commonly caused by trauma, and characterized by a partially erect penis with low pain severity [1,2,3]. Continuous ischemic priapism induces permanent smooth muscle necrosis and fibrosis, which can cause smooth muscle dysfunction in the corpus cavernosum [6]. Stuttering (recurrent) priapism refers to the recurrent and painful erection and shows similar rigidity and severity of pain to that observed with ischemic priapism, but the length of erection time is relatively short [1,2,3,4,5]. Sickle cell disease is reportedly the most common causes of stuttering priapism in humans [1,2,7].

In human medicine, the European Association of Urology (EAU) and the American Urology Association (AUA) have established a diagnosis and distinguished the subtypes [1,2,3]. A thorough medical and trauma history and the onset of clinical signs are essential for the diagnosis and classification of priapism [1,2,5]. Physical examination should include the entire penis and perineal region, and rigidity, color, and scale of pain should be accurately evaluated [1,2,3,4,5]. Laboratory testing including complete blood count, platelet count, and coagulation profile is recommended to detect hematological disorders [5,8,9]. Corporal blood aspiration for gas analysis assists in distinguishing between ischemic and non-ischemic priapism [1,2,3,4]. In ischemic priapism, the blood analyzed during initial corporal aspiration is usually dark in color, and gas analysis shows hypoxia (pO_2_ < 30 mmHg and pCO_2_ > 60 mmHg, and a pH of 7.25) [1,2,3,4,5,10]. Color-flow Doppler ultrasonography of the entire penis and scrotum is a useful tool for distinguishing non-ischemic from ischemic priapism and can provide information on fistula and blood flow in dogs and humans [5,7,11]. For refractory priapism, clinicians can consider penile magnetic resonance imaging (MRI) and corporal biopsies for the diagnosis of corpus cavernosum smooth muscle necrosis [7].

A total of 12 cats have been diagnosed with priapism since 1989; none of these studies reported non-ischemic priapism, and the treatment was mostly related to surgical options such as penile amputation and perineal urethrostomy, due to poor responses to conservative management [12,13,14,15,16,17].

Non-ischemic priapism has been reported in humans and dogs, but not in cats [1,2,3,4,5,6,8,18]. This is the first case report of non-ischemic priapism in a cat. Herein, we describe the difference in the diagnosis and treatment of non-ischemic priapism in cats and in dogs and humans.

## 2. Case Presentation

A 14-year-old male neutered British shorthair cat weighing 4.88 kg was referred to the Veterinary Medicine Teaching Animal Hospital of Chungnam National University with a history of persistent erection, dysuria, and perineum over-grooming for 21 days before admission. Despite 14 days of medical treatment with prednisolone (0.75 mg/kg q12h per os (PO)), metronidazole (5 mg/kg, q12h, PO), and famotidine (0.5 mg/kg, q12h, PO) at a local animal clinic, clinical signs did not improve. On referral to a veterinarian, the patient was bright, alert, and responsive, and blood examination was within the normal range and urethral patency was positive. An Elizabethan collar (EB collar) was used to hinder the over-grooming of the perineum, but the patient showed an immediate obsession with the penis after the removal of the EB collar.

Moreover, the owner complained that the cat looked uncomfortable while urinating, and that the time taken to void urine had increased from 30 s to 2 min with a persistent erection. There was no history of mating, but interestingly, the patient was able to walk accompanied by its owner. Additionally, the patient had a history of cystitis treatment prior to the priapism.

A general physical examination showed a persistent erection, and the penis was erect but not hard enough for penetration. The patient was slightly unsettled in the hospital and reacted aggressively to penis palpation. The surface of the penis was dry, and brown debris was detected at the glans (Figure 1). A soft movable mass (3.1 × 3.2 cm) was detected in the left inguinal area. Body temperature, measured rectally, was 38.9 °C and other vital parameters were within normal limits.

Complete blood count (CBC) was normal except for mild lymphopenia (1.24 × 10^9^ cells/L) (reference interval [RI]: 2–7.2) and mild eosinopenia (0.12 × 10^9^ cells/L) (RI: 0.3–0.7). Serum electrolyte concentration, serum biochemistry, gas analysis, and total T4 revealed no abnormalities except for elevated Feline serum amyloid A (FSAA) level (23 μL/mL) (RI: 0–10) and mild hyperglycemia (171 mg/dL) (RI: 73–134). Urethral catheterization revealed complete patency. Urinary samples collected by ultrasound-guided cystocentesis showed no abnormal laboratory urine test results and a negative result on bacterial culture. Abdominal radiography (XPLRER-900, Medien International Co. Ltd., Anyang, Korea) revealed edema of the perineal region. A cutaneous mass with fat opacity was also detected in the inguinal regions; this was later suspected to be a lipoma (Figure 2). Ultrasonography performed using Philips iU22 SonoCT system (Philips Ultrasound, Bothell, WA, USA) revealed that the erect, swollen penis’s parenchyma was homogenous and hyperechoic. The penile cavernosal artery was not clearly identified by color-flow Doppler ultrasonography with the lowest Nyquist limit range of −2.5–2.5 cm/s (Figure 3). Ultrasound-guided fine needle aspiration of the left inguinal mass identified lipid droplets before staining, and single large clusters of well-differentiated adipocytes with Diff-Quick staining were observed under a microscope.

Despite non-detectable blood flow in corpora cavernosa in color-flow Doppler examination, based on its history and other symptoms (21 days of persistent soft erection without severe pain), the cat was diagnosed with non-ischemic priapism.

Conservative treatment was initiated with flushing of the penis with 0.9% normal saline as the surface was dry and showed brown debris near the glans. Lidocaine gel was used for lubrication and topical anesthesia to reduce mild pain. Topical steroid cream was applied directly to the penis to reduce inflammation and irritation. For medical management, diazepam (0.4 mg/kg, q12h, PO) and gabapentin (10 mg/kg, q12h, PO) were prescribed for sedation and analgesia. With the exception of 10 min of grooming time for the cat, 24-h EB collar wearing was advised to prevent grooming of the perineal region, which could reduce the efficacy of topical ointment and cause further damage. Moreover, to prevent additional exposure and excoriation of the penis, the cat was not allowed to go for a walk. According to the owner, the persistent erection was resolved in 24 h, even though the cat continuously showed a swollen penis within the prepuce. Dysuria gradually improves over 3 days, and palpation-induced pain was resolved in 5 days.

After 7 days, the cat was brought to the animal hospital for physical examination and urogenital ultrasound recheck. FSAA, which showed an elevation at the first presentation, declined to the normal reference range. Physical examination and urogenital ultrasonography revealed significant pain reduction and reduced swelling of the penis. The penis was no longer enlarged or, hard, and it retracted into the prepuce well (Figure 4). According to the owner, the patient was bright and responsive, but seemed less energetic after the prescribed oral medication.

After 10 days of conservative medical management, the cat no longer showed perineal region grooming, and it has not experienced a recurrence during the six-month follow-up period.

## 3. Discussion

The diagnosis of each type of priapism reflects different pathophysiological processes [19,20]. In ischemic priapism, the pressure increases in the corpora cavernosa, which results in acidosis, hypoxia, and necrosis of the smooth muscle, which is a urological emergency [5]. Non-ischemic priapism, also known as a high-flow priapism, is less of a urological emergency, and can be further classified into three types: traumatic, neurogenic, and post-shunting [21]. Non-ischemic priapism is typically associated with unregulated arterial inflow, commonly due to impairment of the cavernsoal artery, and it has been reported in human patients with neurological conditions such as spinal cord injuries and meningomyelocele [5,10,22]. The cause of non-ischemic priapism, in this case, was considered to be idiopathic or traumatic because of the absence of neurological signs or a history of drug ingestion. As the patient could walk, there was a possibility of perineal or penile trauma, which the owner did not recognize. A traumatic cause needed to be considered because a 2–3 week delay between the injury and the occurrence of the priapism is possible [2].

To the best of our knowledge, this is the first report, describing feline non-ischemic priapism. Non-ischemic priapism is rare and is a condition that is unfamiliar, even to many human urologists and andrologists [19,21]. First reported in 1960 by Burt et al. [9], non-ischemic priapism constitutes less than 5% of priapism cases in humans, which is considerably less than ischemic priapism [19].

As priapism can be a urological emergency, it is important for veterinarians to quickly differentiate between the types [19]. The diagnosis of priapism is based on history, physical examination, corporal blood gas analysis, and color-flow ultrasound scanning of the penis and perineum [1,2,3,5]. Computed tomography, magnetic resonance imaging, and angiography are not commonly used for the diagnosis of ischemic priapism but could be performed for the identification of arteriocorporal fistula in non-ischemic priapism [4]. Key components in taking the history of priapism are duration of erection, existence of pain, medication, mating, and trauma of the perineum or penis [1,2,3,4,5,19]. Non-ischemic priapism can be suspected with a history of pain, no progressive discomfort, history of copulation, or trauma to the penis or perineum [1,2,3,4,19]. Physical examination of corpora cavernosa hardness, evidence of trauma to the perineum, and pain severity should be evaluated [23]. Generally, the corpora cavernosa is partially rigid, and pain is not severe in non-ischemic priapism [1,2,3,4,5,19]. Laboratory testing including CBC and coagulation profiling provides baseline information for diagnosing underlying disease, but corporal blood gas analysis is essential for distinguishing non-ischemic from ischemic priapism [1,2,3,4,5,10,19]. According to human guidelines on priapism, the first corporal aspirated blood gas values show pO_2_ > 90 mmHg, pCO_2_ < 40 mmHg and pH 7.35–7.4, as cavernous tissue appears to be well-oxygenated in non-ischemic priapism [1,2,3,4,5]. However, because the size of feline penises is significantly smaller than those of canines and humans, it was impossible to aspirate blood from the caverrnosal bodies for gas analysis in this case [24,25,26,27,28,29].

Color-flow Doppler ultrasound scanning is also an important diagnostic tool that enables clinicians to confirm cavernosal arterial blood flow and the existence of a fistula [19,30]. Sonographic characteristics of priapism have been reported in veterinary medicine [8,18]. A penile Doppler study in ischemic priapism displays absent cavernosal blood flow and low echogenicity resulting from sinusoidal thrombosis [30]. In human, non-ischemic priapism is characterized by normal or raised cavernosal arterial blood flow with high diastolic velocities [30]. In this case, no cavernosal arterial flow was identified, even with a low Nyquist limit (−2.5–2.5 cm/s) in the color flow Doppler study due to the small penis size in the feline. Generally, the absence of cavernosal arterial blood flow is a feature of the ischemic priapism [1,2,3]. However, the vessel’s inner diameter could be the factor that can not be detected by color-flow Doppler ultrasound scanning [31]. Considering anatomical limitations of the feline penis, we speculated that the vessel diameter could have been the reason. Further studies are needed because this case report first described the erected feline’s penile ultrasound.

Conservative treatment and non-surgical management are recommended for non-ischemic priapism which is not an emergency and resolves naturally without any significant side effects [19]. The application of ice and compression to the perineum is generally recommended for conservative treatment [1,2,3,4,5,13,19]. In veterinary medicine, conservative management using analgesia, anti-inflammatory drugs, lubrication of the penis, and prepuce for protection, and the application of EB collars is recommended [13,31]. Blood aspiration from the cavernosal bodies and irrigation have been concluded to be ineffective in the treatment of non-ischemic priapism [19]. Invasive interventions such as selective arterial embolization using autologous clots, gel foam, microcoils, ethylene-vinyl alcohol copolymer (EVOH), N-butyl-cyanoacrylate (NBCA), or ligation of the fistula by surgical management are possible for the management of non-ischemic priapism in humans [2,3,19].

In a review of the literature in human medicine, success rates of first intervention in non-ischemic priapism by blood clots 61.7%, gel foam 75%, EVOH/NBCA 77.3%, microcoil 80%, surgical 90%, and observation/conservative 46.9% have been reported [5]. In this case, our team decided to perform conservative and supportive treatment after thorough discussion with the owner about the risk and complications of invasive interventions and surgical management, as well as the lack of significant results expected from delaying interventions [1,2,3,4,5,19,32]. As the surface of the penis showed dry brown debris, we flushed the penis and perineal region with 0.9% normal saline. At the time of diagnosis, considering the patient’s history of bacterial cystitis, we performed flushing of the urinary bladder and urethra tract with 0.9% normal saline after collecting samples for bacterial culture and antimicrobial sensitivity testing by cystocentesis. After flushing, we applied lidocaine gel and topical steroid ointment (hydrocortisone 1% cream) for lubrication, anti-inflammatory, and analgesic effects. Even though the pain of non-ischemic priapism is not usually severe, in this case, the cat showed signs of pain and displayed over-grooming of the perineal region. We prescribed diazepam (0.4 mg/kg, q12h, POBID) and gabapentin (10 mg/kg, q12h, PO) for analgesia and sedation. Use of an EB collar was strictly indicated to reduce further damage to the penis.

Serum amyloid A (SAA) is a prominent constituent of acute phase proteins (APPs) that are elevated in various diseases [33]. In cats, SAA has been reported as a biomarker for the presence of inflammation and prognostic indicators [34]. Feline serum amyloid is elevated in tumors, infections, inflammation, chronic kidney disease, and traumatic disorders [35,36]. In the case reported here, the cat showed no remarkable findings on complete blood cell count, serum chemistry, and blood gas analysis, but showed a significantly elevated Feline serum amyloid A level (23 ug/mL). After conservative and supportive treatment with topical ointment and analgesia, FSAA decreased from 23 ug/mL to 5 ug/mL with resolution of the patient’s major complaints. The cat in this report showed a high level of serum amyloid A (SAA), which demonstrated the presence of inflammation. The previous history of cystitis was ruled out based on urinalysis and urine culture tests. As the cat persistently showed perineal region overgrooming, tissue damage or unknown inflammation was suspected, affecting the non-ischemic priapism.

A few single reports of non-ischemic priapism have been documented in canine veterinary medicine, but not in feline medicine [8,18]. Priapism is very rare, and a total of 12 cases have been previously described, none of which mentions non-ischemic priapism [12,13,14,15,16,17]. The treatment of priapism was mostly related to surgical options such as penile amputation and perineal urethrostomy, as priapism was not fully resolved by conservative and supportive management [12,13,14,15,16,17].

In this case, the feline patient was uncomfortable but did not seem to be in acute pain, and the erection of the corpora was not completely rigid. In two human medicine reports, the diagnosis of non-ischemic priapism in an infant was based on history and physical examination due to difficulties of blood gas analysis and color-flow Doppler ultrasound scanning [37,38]. Based on its history, physical examination, lack of severe pain, and complete clinical resolution of erection spontaneously after conservative management, the patient’s diagnosis appeared to be non-ischemic priapism. After the analgesia and supportive management prescription, the patient showed no signs of recurrence during the 6-month follow-up.

## 4. Conclusions

To our knowledge, this case describes the first reported case of feline non-ischemic priapism in a cat, a 14-year-old neutered male British Shorthair with a 21-day history of persistent erection and dysuria accompanied by perineal region overgrooming. The cat with non-ischemic priapism showed complete resolution of clinical signs with conservative and supportive management. No complications or side effects were observed during the 6-month follow-up period. Traditional methods of diagnosing non-ischemic priapism are not feasible in all cats due to anatomical limitation of feline penises. Therefore, thorough history taking and physical examinations are crucial in feline non-ischemic priapism.

## Figures and Tables

**Figure 1 vetsci-09-00029-f001:**
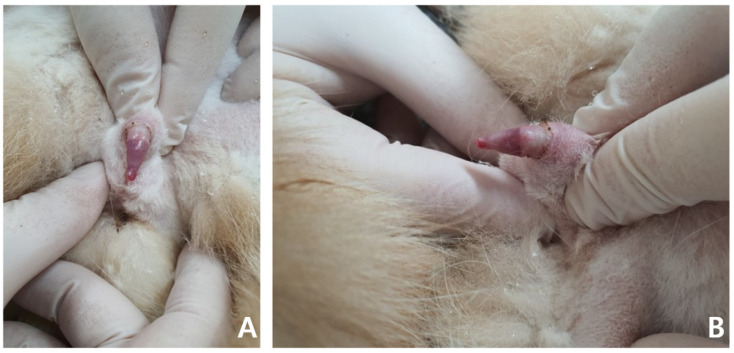
Non-ischemic priapism in a 14-year-old neutered male British shorthair. The penile tip is dry and shows a soft erection (**A**,**B**).

**Figure 2 vetsci-09-00029-f002:**
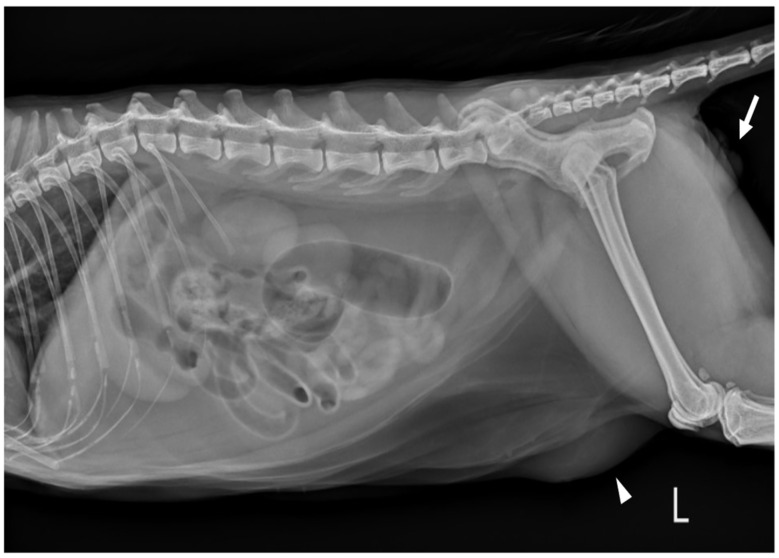
Left lateral abdominal radiograph in a cat revealed edema of the perineal region (white arrows). The cutaneous mass with fat opacity in the inguinal region (white arrowheads) was cytologically suspected as a lipoma.

**Figure 3 vetsci-09-00029-f003:**
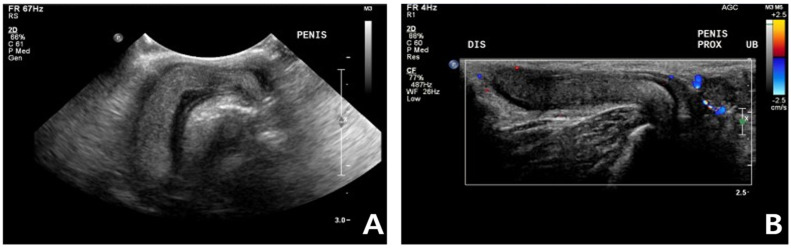
Ultrasonography in a cat with non-ischemic priapism (**A**) swollen penis with homogeneous parenchyma; (**B**) The color-flow Doppler examination did not clearly identify penile cavernosal artery blood flow due to the small size of the feline penis.

**Figure 4 vetsci-09-00029-f004:**
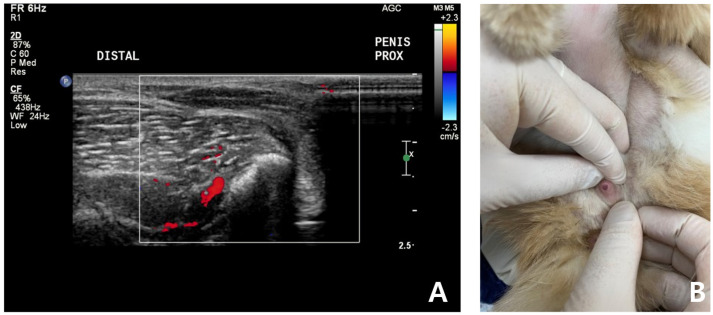
Follow-up ultrasound examination of a cat with non-ischemic priapism after complete clinical resolution of persistent soft erection. No significant penile cavernosal artery blood flow was detected due to the anatomical limitation of feline penis (**A**). Penis was no longer enlarged, not hard and retracted into the prepuce well (**B**).

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
