# Peer review of "Presumptive Non-Ischemic Priapism in a Cat"

_vetsci, 2022, doi:10.3390/vetsci9010029_

Round 1

Reviewer 1 Report

The manuscript presents a case of a non-ischemic priapism in a cat, which has not been reported before. As priapism is vary rare condition in this species, presented approach to this disease may be interesting and useful for veterinary practitioners.

The manuscript is well structured and easy to follow. Numerous photos enrich the publication.

I have only minor remarks:

Line 95 – a typo in ’Ultrasound0guided”

Lines 105-106 – a non-ischemic priapism is characterised by high flow through the cavernosa, lack of the flow in this case is indicating an ischemic priapism. I suggest to change as following:

‘Despite non-detectable blood flow in corpora cavernosa in color-flow Doppler examination, based on its history and other symptoms (21 days of persistent soft erection without severe pain), the cat was diagnosed with non-ischemic priapism.

Lines 166 – 173 – not important details, disturbing the flow. To be removed.

Lines 193 - for the management of non-ischemic priapism IN HUMANS.

Author Response

We would like to thank your thoughtful review of the manuscript. You raise important issues, and your comments helped improve the manuscript. I hope that you will find our responses satisfactory. We are willing to further edit the revised version of the manuscript in response to any additional suggestions you may have. Please see the attachment

Reviewer 2 Report

The manuscript is an interesting case report on a case of Priapism in cat, unfortunately there are some substantial issues that should be improved for the final approval.

  1. “Nonischemic priapism is a disorder of arterial flow that typically presents as a partially erect, nontender erection. The diagnosis of non-ischemic priapism is obtaining a cavernosal blood gas and/or duplex ultrasound. Color duplex ultrasound of the penis and perineum will demonstrate normal to increased flow within the cavernosal arteries. Aspiration of the cavernosum will demonstrated bright red blood and a normal arterial blood gas profile.” [Bassett, J. and Rajfer, J., 2010. Diagnostic and therapeutic options for the management of ischemic and nonischemic priapism. Reviews in urology, 12(1), p.56.]. In this study, the diagnosis of non-ischemic priapism is uncertain, because the ultrasound didn’t reveal how we expected in case of non-ischemic priapism and the cavernosal blood gas analysis were not performed due to small size of the penis. The case report should be redirected in a way toward to characterizing and suggesting the diagnosis. Indeed, I suggest to change the title in “Non- ischemic priapism in a cat?” or something similar.

  1. The ultrasound findings are not explained in detail, and they are not correspondent to the figure 3, they should be improved (lines 92-95 and Figure3). Furthermore, the discussion section dedicated to the color-flow doppler ultrasound (lines 174-182) is not clear and should be improved. I suggest to add “In human,…” at line 178, before “non- ischemic priapism” and to explain better that in this case there is not a classic presentation of doppler during non-ischemic priapism. May should add the references and explanation of why the color doppler flow of cavernosal artery is not identified.

  1. In this study, attempts to identify the possible causes of priapism were not undertaken. “Non-ischemic priapism is typically associated with unregulated arterial inflow, commonly due to impairment of the cavernsoal artery, and it has been reported in human patients with neurological conditions such as spinal cord injuries and meningomyelocele “. Anyway, in the discussion (line139) must be added that those attempts were not performed and the authors should speculate on possible causes of priapism in this case.

  1. The presence of high levels of serum amyloid A (SAA) should be commented, as it should mean a presence of an inflammation process that could be the cause of priapism. If the fibrinogen (chronic phase protein) was also checked at the first evaluation and after 7 days, it will be interesting to comment the possible chronicity of the process.  

However, I have also identified some minor issues that may improve the case reports.

Lines 82-83 please correct “1.24x109” in 1.24 x 10 9” and “0.12 x 109” in “0.12 x 10 9“.

Line 95 please correct “Ultrasound0guided” in “Ultrasound-guided”.

Lines 160-161 please delete the break line.

Line 181 please change “lowest” in “a low”.

Author Response

(The authors gave the same response as above.)

Reviewer 3 Report

Reviewer comments for manuscript ID vetsci-1457027 entitled ‘Non-Ischemic Priapism in a Cat’

General comments

An interesting and rare case presented. The manuscript has been very nicely written with schematic steps to arrive at a diagnosis. I appreciate the authors for a thorough professional write up following a meticulous treatment plan. The introduction attracts the attention of a reader to read the whole manuscript. Discussion is logical, thorough but precise. Frankly, I found no errors in the manuscript. I highly recommend the publication of this clinical work.

Specific comments

Line 70; Please delete ‘a feline’ and ‘and ‘

Line 71: Please clarify this line ‘3 and 6 years’

Lines 86-87: Please reframe ‘Urethral catheter insertion was performed, which revealed a positive urethral patency’ as ‘Urethral catheterization revealed complete patency’

Lines 87-89: Was cystocentesis performed under ultrasound guidance? Please clarify.

Author Response

(The authors gave the same response as above.)

Reviewer 4 Report

The manuscript is an interesting case report on a case of Priapism in cat that could give important new useful information.

The clinical case is correctly presented but the analyses have not been performed with the highest technical standards or have not been described in sufficient detail.

  • The definition of ischemic priaprism related to the blood flow of the penis (arterial - high flow or veno-occlusive, low flow) should be clarified because is contradictory in several points (see line 22 and 24 eg, and in the discussions) as well as with the bibliography.
  • In order to make a diagnosis of ischemic or non-ischemic priapism in human (2017 doi: 0.5152/tud.2017.59458  Ischemic priapism: A clinical review Joanne Ridgley,1 Nicholas Raison,2 M. Iqbal Sheikh,3 Prokar Dasgupta,2 M. Shamim Khan,2 and Kamran Ahmed2)  and in cat (Priapism after castration in a cat. Swalec KM, Smeak DD. J Am Vet Med Assoc. 1989 Oct 1;195(7):963-4) beyond the clinical evidence given by the clinical examination different diagnostic tests can be used: cavernosal blood gas analysis, eco color doppler examination of the cavernosal blood flow, penis histological examination. 

In this manuscript the cavernosal blood gas analysis was not performed due to small size of      the penis and the ultrasound examination is not well described: there are contradictions between the legend of figure 4, results and discussion.

Author Response

(The authors gave the same response as above.)

Round 2

Reviewer 2 Report

The manuscript has been improved after major revisions, and now it can be accepted in present form